# Circulating FGF-21 as a Disease-Modifying Factor Associated with Distinct Symptoms and Cognitive Profiles in Myalgic Encephalomyelitis and Fibromyalgia

**DOI:** 10.3390/ijms26167670

**Published:** 2025-08-08

**Authors:** Ghazaleh Azimi, Wesam Elremaly, Mohamed Elbakry, Anita Franco, Christian Godbout, Alain Moreau

**Affiliations:** 1Viscogliosi Laboratory in Molecular Genetics of Musculoskeletal Diseases, Office 2.17.027, Azrieli Research Center, CHU Sainte-Justine, 3175 Cote-Ste-Catherine Road, Montreal, QC H3T 1C5, Canada; 2Department of Biochemistry and Molecular Medicine, Faculty of Medicine, Université de Montréal, Montreal, QC H3T 1J4, Canada; 3Open Medicine Foundation ME/CFS Collaborative Center, CHU Sainte-Justine, Montreal, QC H3T 1J4, Canada; 4ICanCME Research Network, Azrieli Research Center, CHU Sainte-Justine, Montreal, QC H3T 1C5, Canada; 5Biochemistry Division, Chemistry Department, Faculty of Science, Tanta University, Tanta 31527, GG, Egypt; 6Department of Stomatology, Faculty of Dentistry, Université de Montréal, Montreal, QC H3T 1J4, Canada

**Keywords:** myalgic encephalomyelitis, fibromyalgia, FGF-21, biomarkers, cognition, symptoms

## Abstract

Myalgic encephalomyelitis (ME) and fibromyalgia (FM) are overlapping syndromes characterized by persistent fatigue, cognitive difficulties, and post-exertional malaise (PEM), yet they lack objective biomarkers for diagnosis and treatment. Fibroblast growth factor 21 (FGF-21), a stress-responsive metabolic hormone, may offer a promising avenue to distinguish subtypes within these patient populations. In this cross-sectional study, plasma FGF-21 levels were measured in 250 patients (FM = 47; ME = 99; ME + FM = 104) and 54 healthy controls. Participants were categorized based on FGF-21 levels into three groups: low (0–50 pg/mL), normal (51–200 pg/mL), and high (>200 pg/mL). Symptoms burden and cognitive function were assessed using validated questionnaires (SF-36, MFI-20, DSQ, DPEMQ) and the BrainCheck platform. A standardized mechanical provocation maneuver was used to induce PEM. Results showed that elevated FGF-21 levels were frequently observed in ME and ME + FM but varied widely across all groups. Stratification by circulating FGF-21 levels, rather than diagnosis alone, revealed distinct symptom and cognitive profiles. Low FGF-21 levels were linked to worsened PEM perception in FM, increased PEM severity and immune/autonomic symptoms in ME, and poorer mental health in ME + FM. Conversely, high FGF-21 levels correlated with better cognition in ME but greater fatigue in ME + FM. These findings suggest that FGF-21 may serve as a valuable biomarker for identifying clinically meaningful subtypes within ME and FM, supporting the development of personalized treatments. Furthermore, discrepancies between DSQ and DPEMQ highlight the need for objective PEM assessment tools. Overall, FGF-21 shows potential as a biomarker to guide precision medicine in these complex conditions.

## 1. Introduction

Myalgic encephalomyelitis (ME) and fibromyalgia (FM) are chronic, debilitating conditions that share overlapping clinical features, including profound fatigue, pain, cognitive impairment, sleep disturbances, and autonomic dysfunction [1,2,3,4]. While ME is clinically defined by the hallmark symptoms of post-exertional malaise (PEM), fatigue, cognitive impairment, dysautonomia and unrefreshing sleep [5], FM is primarily characterized by chronic widespread musculoskeletal pain, fibro-fog and fatigue [6,7]. Together, these conditions affect a significant portion of the population, ranging from 0.4 to 2.5% for ME [8] and 2 to 4% for FM [9], and impose a substantial burden on individuals and healthcare systems alike.

The diagnostic complexity of ME and FM arises from their heterogeneous clinical manifestations, reliance on self-reported symptoms, and the absence of objective biomarkers [10,11,12]. Standardized diagnostic criteria, such as the Canadian Consensus Criteria for ME [13] and the American College of Rheumatology (ACR) criteria for FM [14], are useful but limited, especially in cases where both syndromes co-occur [15]. Despite ongoing research, the pathophysiological mechanisms of both conditions remain poorly defined, although evidence supports roles for immune dysregulation [16], neuroinflammation and autonomic dysfunction [3,17], as well as mitochondrial abnormalities [18]. The lack of peripheral biomarkers continues to hinder efforts in disease stratification, diagnosis, and the development of targeted therapies [3].

Fibroblast growth factor 21 (FGF-21) is a stress-inducible metabolic hormone with pleiotropic roles in energy regulation, mitochondrial function, and inflammation [19,20,21,22,23,24,25,26,27]. Paradoxically, FGF-21 is elevated in various pathological conditions such as obesity, type 2 diabetes, and non-alcoholic fatty liver disease [28,29,30,31,32,33,34], a phenomenon often interpreted as FGF-21 resistance or a compensatory response to chronic metabolic stress [35]. Emerging evidence also suggests that FGF-21 dynamics, including baseline levels and physiological responses, may differ by sex in various contexts [36,37,38]. Given its associations with metabolic dysfunction and its emerging role as a biomarker of mitochondrial stress, FGF-21 is a promising candidate for investigation in ME and FM. While preliminary data suggest its involvement in ME [39], the utility of this approach in stratifying clinical subtypes across ME, FM, and their comorbid presentations remains unexplored. Despite extensive research into FGF-21 in metabolic and liver pathologies [19,20,21,22,23,24,25,26,27,28,29,30,31,32,33,34,35], only a handful of smaller studies have examined its specific relevance to ME, producing preliminary and sometimes inconsistent findings. Some reports indicate elevated FGF-21 levels in ME patients [39], but the correlation with symptom severity remains unclear. Another study has identified higher FGF-21 levels in a distinct subgroup of ME patients characterized by a unique metabolic phenotype, particularly low serum levels of fatty acid derivatives [5]. Such discrepancies likely stem from limitations like smaller cohort sizes, heterogeneous patient populations lacking rigorous diagnostic sub-stratification, and insufficient characterization of the full clinical spectrum of ME and FM, especially the core symptom of PEM. These gaps underscore the need for a more comprehensive examination of how FGF-21 contributes to the pathophysiology and phenotypic variability in ME and FM, as well as its potential as a biomarker for patient stratification. This study aims to characterize circulating FGF-21 levels in these patient groups and to evaluate their associations with symptom severity and cognitive function. By doing so, it aims to provide a novel framework for biomarker-guided stratification and support the development of personalized management strategies for these complex and often overlapping conditions.

## 2. Results

### 2.1. Participant Characteristics and Questionnaire Validation

A total of 304 individuals were included in the analysis: 250 patients divided into FM (*n* = 47), ME (*n* = 99), and ME + FM (*n* = 104) groups, alongside 54 sedentary healthy controls (HC). Demographic characteristics were comparable across groups, with no significant differences in age or BMI (mean age ~47–48 years; mean BMI ~25–26 kg/m^2^). As expected, all patient groups exhibited a strong female predominance, consistent with disease prevalence trends: FM (38 females/9 males), ME (83 females/16 males), and ME + FM (94 females/10 males), while the HC group was balanced (25 females/29 males). The average illness duration was 11 ± 1.6 years (FM), 12 ± 1.1 years (ME), and 14 ± 1.3 years (ME + FM) (Table 1).

Self-reported health and symptom burden were assessed using the SF-36, MFI-20, and DSQ instruments. SF-36 scores, where higher values indicate better health, were significantly reduced in all patient groups compared to HC (*p* < 0.0001), confirming impaired general health. MFI-20 and DSQ scores, where higher values denote greater symptom severity, were significantly elevated in patients relative to controls (*p* < 0.0001). No significant differences were observed between FM, ME, and ME + FM groups across these instruments, indicating comparable levels of fatigue and symptom burden (Figure 1).

### 2.2. Circulating FGF-21 Levels Reveal Clinically Relevant Subtypes Across ME, FM, and ME + FM

Plasma FGF-21 concentrations (pg/mL) were significantly elevated in patients with ME and in those with FM as a comorbidity (ME + FM), compared to healthy controls (Figure 2a). In contrast, no significant elevation was observed in patients diagnosed with FM alone. These findings suggest that elevated plasma FGF-21 levels are more specifically associated with ME pathology, whether isolated or co-occurring with FM. Across all groups, including sedentary healthy controls, circulating FGF-21 levels exhibited substantial inter-individual variability, ranging from very low (<50 pg/mL) to markedly high concentrations (>200 pg/mL).

We also examined whether biological sex influenced FGF-21 concentrations. As shown in Figure 2b, no statistically significant differences were found between female and male participants within the FM, ME, or ME + FM groups, nor among healthy controls. Given the higher proportion of female participants in our cohort (Table 1), which aligns with the known epidemiology of these conditions [4], this finding strengthens the interpretation that elevated FGF-21 is a feature of disease pathology itself, independent of sex.

### 2.3. Differential Clinical Consequences of Low Circulating FGF-21 Across FM, ME, and ME + FM

To explore the clinical implications of low circulating FGF-21 concentrations (0–50 pg/mL), we performed subgroup analyses across diagnostic groups (FM, ME, ME + FM), examining correlations with symptom severity, cognitive performance, and mental health. While low FGF-21 did not associate with symptoms in healthy controls, distinct and disease-specific associations emerged in patient subgroups (Table 2).

### 2.4. Low FGF-21 Is Associated with Acute Exertional Sensitivity and Cognitive Rigidity in FM

Among FM patients, low FGF-21 levels were significantly associated with greater symptom severity following exertion, as assessed by the DSQ. These associations were strongest for items reflecting immediate physical and mental fatigue after minimal effort, consistent with the sensory-perceptual hypersensitivity often observed in FM (Table 3). In contrast, no significant associations were found between FGF-21 and PEM when assessed using the DPEMQ following a standardized 90 min passive exercise protocol (Table 4). This discrepancy underscores a key distinction in FM: PEM manifestations appear more related to acute perceptual reactivity rather than the delayed, systemic crashes typically seen in ME (Table 3 and Table 4). Supporting this, strong negative correlations were identified between FGF-21 levels and specific DSQ PEM items, including “mentally tired after the slightest effort” (r = −0.89, *p* = 0.003), “minimum exercise makes you physically tired” (r = −0.83, *p* = 0.012), and overall PEM severity (r = −0.90, *p* = 0.005) (Table 3). However, no significant correlations were found between circulating FGF-21 levels and DPEMQ scores (r = −0.44, *p* = 0.38) (Table 2) or its individual items (Table 4), which reinforces the view that classical PEM is not a characteristic of FM. By “classical PEM”, we refer to the delayed, disproportionate worsening of symptoms following exertion that is typically observed in conditions like ME, including both physical and cognitive crashes that can last for days and are not alleviated by rest.

Objective assessment of cognitive functions via BrainCheck at baseline revealed that FM participants with low circulating FGF-21 levels exhibited significantly lower scores in mental flexibility compared to those in the normal and high FGF-21 groups (Figure 3). However, no statistically significant correlations were found between FGF-21 levels and overall cognitive performance across the whole FM group (BrainCheck combined: r = 0.12, *p* = 0.45; Table 5).

### 2.5. Low FGF-21 Reflects Delayed PEM and Neuroimmune Vulnerability in ME and ME + FM

In patients with ME, low circulating FGF-21 levels were strongly associated with increased post-exertional symptom severity and systemic dysfunction. Specifically, FGF-21 showed significant negative correlations with autonomic/endocrine/immune symptoms (DSQ: r = −0.52, *p* = 0.03) and with overall PEM severity measured by the DPEMQ following a standardized passive mechanical challenge (r = −0.65, *p* = 0.03). These associations were especially pronounced for items such as “flu-like symptoms after exertion” (r = −0.76, *p* = 0.007; Table 4), reflecting the characteristic delayed, multi-system response to exertion in ME. In contrast, no significant association was found between low FGF-21 levels and PEM severity using the broader, more subjective DSQ measure (r = −0.001, *p* = 0.97; Table 3), highlighting the importance of using condition-specific tools such as the DPEMQ to capture the exertional dynamics unique to ME.

Cognitive scores from the DSQ (r = −0.12, *p* = 0.64) and mental health ratings from the SF-36 (r = −0.42, *p* = 0.097) were not significantly correlated with FGF-21 in ME. However, objective cognitive testing using BrainCheck revealed that ME patients with low plasma FGF-21 had significantly lower scores in immediate memory recognition compared to those in the normal and high FGF-21 groups (Figure 3d). Across the whole ME cohort, FGF-21 levels showed a positive correlation with the BrainCheck combined percentile (r = 0.26, *p* = 0.043), immediate memory recognition (r = 0.39, *p* = 0.002), and delayed memory recognition (r = 0.33, *p* = 0.009; Table 5).

In patients with comorbid ME and FM (ME + FM), low circulating FGF-21 levels were associated with a more complex and multidimensional clinical profile. In this group, lower plasma FGF-21 levels were significantly associated with greater cognitive dysfunction (DSQ cognitive score: r = −0.67, *p* = 0.03) and poorer self-perceived mental health (SF-36 mental score: r = 0.68, *p* = 0.03) (Table 2). These findings suggest an amplification of FGF-21-related neurocognitive deficits in the presence of overlapping symptom domains. Although not statistically significant, moderate negative correlations were also observed with autonomic/endocrine/immune symptoms (DSQ: r = −0.61, *p* = 0.062) and PEM severity (DSQ: r = −0.39, *p* = 0.26), suggesting a broader but less robust systemic impact of low FGF-21 in this subgroup. No significant differences in BrainCheck cognitive domains were observed across FGF-21 strata within this group, and correlation values remained weak and non-significant (Table 5).

Importantly, the clinical impact of low circulating FGF-21 appears to be disease-specific. Among healthy individuals with similarly low FGF-21 levels, no significant correlations were observed between FGF-21 and fatigue or symptom burden (Table 2). Furthermore, across all diagnostic groups, participants with plasma FGF-21 levels in the normal range (51–200 pg/mL) showed no significant associations between FGF-21 and symptom severity.

### 2.6. High Circulating FGF-21 Levels Show Divergent Clinical Significance Across Diagnoses

To examine the clinical implications of elevated circulating FGF-21 concentrations (>200 pg/mL), we examined group-specific associations with symptom dimensions, fatigue subscales, and cognitive performance across participants with ME, FM, ME + FM, and healthy control. Fatigue severity was evaluated using the MFI-20 questionnaire, and broader symptomatology was assessed with the DSQ and the SF-36 questionnaires. Cognitive performance was evaluated through BrainCheck assessments.

Among ME patients with high circulating FGF-21 levels, a significant negative correlation was found between plasma FGF-21 concentrations and MFI-20 mental fatigue scores (r = −0.53, *p* = 0.004; Table 6). No significant correlations were observed with physical fatigue, cognitive DSQ scores, DSQ PEM, or SF-36 mental or physical health scores. BrainCheck assessments showed higher scores in immediate memory, delayed memory, and global cognitive performance in this subgroup (Figure 3).

In ME + FM patients with high plasma FGF-21 levels, a significant positive correlation was found between FGF-21 and MFI-20 physical fatigue (r = 0.37, *p* = 0.02; Table 6). No significant correlations were observed with MFI-20 mental fatigue (r = −0.02, *p* = 0.89). Furthermore, FGF-21 levels did not correlate with broader symptomology assessed by the DSQ, including cognitive DSQ scores, DSQ PEM, or overall perceived health as captured by SF-36 mental or physical health scores. BrainCheck cognitive performance scores in this subgroup did not differ significantly from other FGF-21 strata.

Among FM patients with high circulating FGF-21, no statistically significant associations were observed between FGF-21 concentrations and MFI-20 mental fatigue (r = 0.13, *p* = 0.63) or physical fatigue (r = 0.14, *p* = 0.61) (Table 6). Similarly, no associations were observed with DSQ cognitive scores, DSQ PEM, or SF-36 scores. BrainCheck cognitive data in this subgroup showed no significant variation across FGF-21 levels (Figure 3). Similarly, among healthy control participants with high circulating FGF-21 levels, no significant correlations were found between FGF-21 levels and MFI-20 mental fatigue (r = 0.13, *p* = 0.68) or physical fatigue (r = 0.21, *p* = 0.50; Table 6). No associations were found with DSQ PEM or cognitive domains, or with SF-36 scores.

## 3. Discussion

This study provides critical insights into the multifaceted role of circulating FGF-21 in ME, FM, and their comorbid presentation (ME + FM), demonstrating its complexity as a disease-modifying biomarker that varies not only with concentration but also across phenotypic subgroups. Our comprehensive characterization of patient cohorts, augmented by a miRNA signature test [40], enabled a rigorous examination of these relationships. Consistently, elevated circulating FGF-21 levels in ME and ME + FM, relative to healthy controls, support hypotheses implicating metabolic dysfunction, mitochondrial impairment, and chronic systemic stress as central to the pathophysiology in these conditions [41,42]. Indeed, our results corroborate and extend prior observations of elevated FGF-21 in subsets of ME patients [5,39] and other chronic, fatiguing, or inflammatory illnesses [43,44], reinforcing its potential as a biomarker of systemic metabolic stress in these disorders. Elevation of circulating FGF-21 levels may be viewed as a compensatory response to cellular stressors [45] and appear to be strongly involved in these diseases. Importantly, despite the well-documented female predominance in ME and FM patient populations, this observed elevation in FGF-21 was found to be independent of biological sex in our cohorts. This finding crucially distinguishes itself from the broader literature, which frequently reports significant sex-dependent differences in FGF-21 baseline levels [36,37] and physiological actions across various other metabolic and stress-related diseases [38]. This independence of sex in ME and ME + FM suggests that the underlying mechanisms driving FGF-21 elevation may operate through pathways that transcend typical sex-specific responses, highlighting a potentially unique aspect of their pathophysiology.

A key strength of this study is the stratification of participants by circulating FGF-21 levels, which uncovered phenotype-specific symptom correlations. In FM patients, low circulating FGF-21 was linked to stronger associations with retrospective PEM (DSQ), but not with the DPEMQ, following a standardized 90 min passive exercise. This likely reflects differences in how these tools capture symptomatology. The DSQ relies on retrospective self-reporting and may be more sensitive to subjective symptom amplification or chronic fatigue perception influenced by central sensitization and emotional factors commonly associated with FM [46,47]. In contrast, the DPEMQ measures symptom changes in response to a controlled provocation, offering a more objective but narrower assessment window. This divergence suggests that low plasma FGF-21 levels may influence broader physiological or perceptual processes that are more evident in daily symptom patterns than in standardized test settings. Meanwhile, in ME patients, the consistent correlation between low circulating FGF-21 levels and both self-reported and provoked PEM, as well as with autonomic and immune symptoms, points to a more systemic metabolic or inflammatory dysfunction, in line with emerging evidence that FGF-21 plays a role in mitochondrial energy regulation and stress adaptation. As FGF-21 typically rises with physical activity [48,49], many patients with severe PEM are largely inactive, possibly lacking this adaptive response. Our findings suggest that insufficient FGF-21 may reflect or contribute to impaired metabolic resilience. In ME + FM patients, low circulating FGF-21 levels were also associated with worse cognitive outcomes, a finding consistent with reports of heightened symptom burden and cognitive deficits in comorbid ME + FM populations [50]. These results challenge the assumption that FGF-21 is consistently elevated in disease, instead indicating that deficient levels may mark more severe symptom expression in specific subgroups.

In contrast, disease groups with high plasma FGF-21 levels presented a complex and often paradoxical picture, underscoring the dual nature of FGF-21, a phenomenon also observed in other chronic conditions where FGF-21 resistance or a state of metabolic overload occurs [51,52]. In ME patients with high plasma FGF-21 levels, higher levels correlated with less severe mental fatigue. This inverse relationship could indicate a successful compensatory mechanism where elevated FGF-21 plays a protective or adaptive role against mental fatigue in the context of ME, a hypothesis supported by preclinical studies on FGF-21’s effects on brain metabolism and neuronal function [53,54,55,56,57]. However, this seemingly beneficial association was not universal. Within the ME + FM group, with high plasma FGF-21 levels, higher levels were associated with more severe physical fatigue. This finding suggests that chronic and highly elevated FGF-21 levels in the comorbid ME + FM group might reflect an overwhelmed system, a state of FGF-21 resistance, or even a maladaptive response contributing to exacerbated physical impairment. This emphasizes that while FGF-21 is responsive to physiological stress [58], its impact on specific fatigue domains is highly context-dependent and may signify different underlying pathological states across distinct patient phenotypes. The specificity of these high-FGF-21 correlations to the MFI-20 fatigue dimensions, rather than broader DSQ symptomology or SF-36 general health, further suggests a targeted involvement in specific energetic or metabolic pathways related to fatigue, rather than a global influence on overall health perception, aligning with the notion of symptom-specific bioenergetic deficits in these conditions [59]. Table 7 presents a comprehensive summary of research investigating FGF-21 in ME, FM and other related chronic conditions, highlighting its expression patterns and associations with symptom severity and disease pathophysiology.

FGF-21 has been increasingly recognized for its potential neuroprotective properties, with evidence suggesting roles in reducing oxidative stress, enhancing mitochondrial function, and supporting neural resilience [53,58]. This neuroprotective aspect of FGF-21 appears particularly relevant to ME. While cognitive impairment was observed across all patient groups, ME patients showed better performance than ME + FM in overall cognition and executive function [60]. More importantly, only in the ME group did higher plasma FGF-21 levels correlate positively with cognitive outcomes, including overall cognitive scores and immediate memory recognition. This specific association suggests that in ME, FGF-21 may exert a beneficial effect on brain function, potentially serving as a buffer against cognitive decline. This finding warrants further exploration but aligns with emerging data on the influence of FGF-21 on neurobiology [53] and specific cognitive deficits reported in ME [60]. The absence of similar correlations in FM and ME + FM groups may reflect different pathophysiological mechanisms or more complex, overlapping disease processes that diminish FGF-21’s protective influence. Collectively, these findings highlight the vast heterogeneity within ME and FM. FGF-21 serves not only as a biomarker but also as a dynamic indicator of metabolic resilience or vulnerability, depending on the clinical phenotype and systemic stress response. This stratification offers a vital framework to move beyond binary diagnoses toward precision medicine interventions.

Therapeutic strategies for individuals with low circulating FGF-21 and accompanying cognitive impairment or PEM should focus on enhancing mitochondrial resilience and amplifying FGF-21 signaling. One readily available pharmacologic option is metformin. A 2024 randomized, placebo-controlled trial in type 2 diabetes patients (1000 mg twice daily for 12 weeks) found that, while circulating FGF-21 levels remained unchanged, metformin significantly suppressed fibroblast activation protein (FAP) activity and upregulated FGFR1c and β-klotho expression in adipose tissue. Notably, FAP is a serine protease that degrades circulating FGF-21, so its inhibition helps preserve FGF-21 bioavailability and enhances its physiological activity [61]. Together, these changes enhance downstream FGF-21 signaling, indicating improved tissue sensitivity to endogenous FGF-21 [62]. These findings support metformin as a candidate therapy for those with impaired FGF-21 dynamics, even if absolute hormone levels are not increased, aligning with its known benefits on AMPK activation and its anti-inflammatory effects (reducing NLRP3, IL-1β, IL-18) [63,64,65]. Additionally, emerging therapies specifically targeting the FGF-21 axis are currently being tested. Efruxifermin (formerly AKR-001), a long-acting FGF-21 analogue, has shown efficacy in phase 2b trials for nonalcoholic steatohepatitis (NASH) with moderate fibrosis (F2/F3), achieving histologic resolution and fibrosis reduction over 24 weeks [66]. Another analogue, pegozafermin (BIO89-100), has demonstrated favorable effects on liver fat and fibrosis in recent trials, including phase 2 studies for NASH and hypertriglyceridemia [67]. Studies in both rodents and humans consistently show that sodium–glucose cotransporter-2 (SGLT2) inhibitors such as dapagliflozin, canagliflozin, and empagliflozin can upregulate hepatic FGF-21 expression and increase plasma FGF-21 concentrations [68]. These agents offer a more direct means of augmenting FGF-21 activity. On a supplement-based front, glucosamine has recently garnered mechanistic support as an inducer of hepatic FGF-21 expression. A 2025 study in hepatocyte models revealed dose- and time-dependent increases in FGF-21 mRNA and protein mediated via activation of the Akt/mTOR/p70S6K axis with an essential dependence on PGC-1α [69]. These cellular mechanisms resonate with glucosamine’s known action on PGC-1α in other tissue contexts [69]. Although promising, the propensity of glucosamine to impair insulin sensitivity mandates careful clinical evaluation, especially in insulin-resistant individuals.

Conversely, patients in the high FGF-21 subgroup within the ME + FM cohort, who exhibit pronounced cognitive impairment and elevated physical fatigue, may be experiencing a state of FGF-21 resistance, wherein elevated circulating levels reflect reduced tissue responsiveness rather than enhanced signaling. In such cases, therapeutic goals should focus on restoring sensitivity to FGF-21 and, where appropriate, reducing its compensatory overproduction. Nutritional interventions remain among the most promising strategies. Clinical and preclinical studies have shown that low-carbohydrate, high-protein diets can reduce plasma FGF-21 levels by improving metabolic efficiency and decreasing hepatic FGF-21 secretion. At the same time, avoiding protein restriction is essential, as insufficient protein intake directly induces FGF-21 expression [70,71,72]. In cases where FGF-21 resistance is evident, enhancing receptor responsiveness becomes a critical therapeutic goal. A promising strategy involves upregulating the expression or activity of β-klotho, the essential co-receptor required for FGF-21 signaling through FGFR1c and FGFR3c [73]. While physical activity is a well-established inducer of β-klotho, this intervention is often impractical for patients experiencing PEM. As an alternative, pharmacologic agents such as peroxisome proliferator-activated receptor gamma (PPARγ) agonists, like rosiglitazone, have demonstrated efficacy in upregulating β-klotho and restoring FGF-21 responsiveness in metabolically stressed tissues [74]. Beyond receptor sensitization, more advanced interventions include synthetic FGF-21 mimetics or receptor agonists. These agents include bispecific proteins and monoclonal antibodies, such as mimAb1 and Avimer C3201, that directly activate the FGFR1/β-klotho complex, thereby replicating the metabolic effects of FGF-21 without requiring elevated endogenous hormone levels [75]. These targeted therapies offer the potential to bypass upstream resistance mechanisms entirely. Emerging evidence also points to the role of senolytic therapies, which selectively eliminate senescent cells. By reducing chronic inflammation and tissue stress, senolytics may indirectly restore β-klotho expression and enhance systemic metabolic signaling. Similarly, anti-inflammatory agents, such as IL-1β inhibitors or NF-κB pathway modulators, may improve receptor sensitivity by mitigating the inflammatory environment [76] that contributes to FGF-21 resistance. Collectively, these therapeutic strategies, ranging from receptor sensitization via PPARγ agonists to direct receptor activation through biologics to systemic modulation with senolytics and anti-inflammatory agents, represent a multifaceted therapeutic framework. Such an approach is especially valuable for patients with elevated circulating FGF-21 and suspected hormone resistance, especially when conventional interventions like exercise are contraindicated due to PEM.

This study offers several notable strengths, including large and well-characterized cohorts, rigorous miRNA-based diagnostic confirmation, standardized exertional stress testing, and a comprehensive, multi-modal assessment approach that surpasses prior single-dimensional biomarker investigations. Nonetheless, certain limitations should be acknowledged. As a cross-sectional analysis, the study cannot establish causal relationships or prognostic implications. However, the robustness and scale of the dataset provide a strong foundation for replication and future hypothesis generation. Notably, this study represents the largest investigation to date of circulating FGF-21 in ME and FM, and the first to systematically compare FGF-21 levels across ME, FM, ME + FM, and healthy controls within a unified analytic framework. The incorporation of objective stress paradigms and cognitive profiling enhances the interpretive resolution of FGF-21 signal variations across clinically relevant subgroups. While independent validation in external cohorts remains the gold standard, our findings serve a critical confirmatory role by replicating and extending prior observations in a significantly larger and more comprehensively assessed population. Future prospective studies in more diverse and multi-center cohorts will be essential to validate the clinical specificity and utility of FGF-21 in stratifying ME and FM phenotypes and to assess its association with underlying mitochondrial signaling dynamics.

In summary, this study establishes that circulating FGF-21 stands out as a promising and nuanced biomarker in the context of ME and FM. Its ability to stratify patients based on symptom severity and cognitive profiles highlights its potential for guiding personalized diagnostic and therapeutic strategies. These findings underscore the importance of integrating FGF-21 into broader biomarker panels to refine disease subtyping and treatment targeting. To fully realize its clinical utility, future research should prioritize longitudinal studies and interventional trials aimed at determining whether modulation of FGF-21 signaling can meaningfully improve patient outcomes.

## 4. Materials and Methods

### 4.1. Study Design and Ethical Approval

This cross-sectional study was conducted in a single center, in accordance with ethical guidelines for human research. The study protocol was approved by the Institutional Ethics Review Board of CHU Sainte-Justine (protocol 4047), and written informed consent was obtained from all participants prior to enrollment.

### 4.2. Study Population

This cross-sectional study involved 304 individuals enrolled initially in ongoing prospective cohorts. All participants were of Caucasian European descent and aged between 18 and 78 years. The cohort included 250 patients and 54 sedentary healthy controls. Patients were initially recruited based on fulfilling the Canadian Consensus Criteria (CCC) for ME. To enhance diagnostic precision and resolve overlapping symptom presentations, all patient classifications were subsequently re-confirmed using a validated diagnostic microRNA (miRNA) panel of 11 circulating miRNAs with a differential expression signature [40]. This molecular profiling allowed confirmation and refinement of diagnostic groupings by aligning clinical phenotypes with disease-specific miRNA expression patterns. Based on the test, individuals were categorized into three diagnostic groups: FM (*n* = 47), ME (*n* = 99), or comorbid ME with FM (ME + FM; *n* = 104), reflecting those who fulfilled both sets of criteria. Sedentary healthy control participants (*n* = 54), defined as engaging in less than two hours of moderate-intensity exercise per week, were frequency-matched to the disease groups based on age and sex, and included based on the absence of chronic fatigue, widespread pain, or any chronic systemic conditions. Individuals with a family history of ME, FM or multiple sclerosis (MS) were also excluded. All participants underwent comprehensive clinical, cognitive, and biomarker assessments as described below.

### 4.3. Post-Exertional Malaise Provocation and Symptom Assessment

To evaluate PEM in a standardized setting, participants underwent a validated passive stress protocol initially designed for ME patients [77]. This involved 90 min of mechanical stimulation using an ABR therapeutic massager with pulsatile compression cuffs applied to one arm (0–4 psi at 0.006 Hz). PEM severity was subsequently quantified five days later, using the DePaul Post-Exertional Malaise Questionnaire (DPEMQ) [78,79].

### 4.4. Blood Collection and Circulating FGF-21 Quantification

Venous blood samples were collected in 6.0 mL K_2_-EDTA vacutainer tubes (BD, Frankin Lakes, NJ, USA). Plasma was isolated by centrifugation at 216× *g* for 10 min at room temperature, aliquoted, and stored at −80 °C until analysis. Plasma FGF-21 concentrations were measured using a quantitative sandwich ELISA (Quantikine^®^ Human FGF-21, R&D Systems, Minneapolis, MN, USA), performed in duplicate. Results were expressed in picograms per milliliter (pg/mL).

### 4.5. Symptom and Health Status Evaluation

Participants completed a series of validated health and symptom questionnaires to assess a broad range of physical and psychological domains. General physical and mental health status was measured using the SF-36, where higher scores indicate better health outcomes [80]. Fatigue was evaluated with the Multidimensional Fatigue Inventory (MFI-20), which assesses five distinct dimensions: general fatigue, physical fatigue, reduced activity, reduced motivation, and mental fatigue. In this measure, higher scores indicate greater severity of fatigue [81]. Additionally, the DePaul Symptom Questionnaire (DSQ) was used to capture symptoms across autonomic, endocrine, immune, cognitive, sleep, and post-exertional malaise (PEM) domains, with higher scores indicating increased symptom severity [77,82]. To measure the PEM after the stress test, we used the DPEMQ questionnaires [78,79].

### 4.6. FGF-21 Stratification

Based on both literature-derived thresholds and the observed distributions in our cohort, plasma FGF-21 concentrations were categorized into three subgroups: low (0–50 pg/mL), normal (51–200 pg/mL), and high (>200 pg/mL) circulating levels [83,84]. These thresholds were uniformly applied across patient and control groups to identify concentration-dependent clinical phenotypes.

### 4.7. Cognitive Assessment at Baseline

Neurocognitive function was assessed at baseline using BrainCheck (BrainCheck, Inc., Austin, TX, USA), a web-based platform consisting of a 15 min test battery. The BrainCheck test was administered to FM (*n* = 39), ME (*n* = 62), and ME + FM (*n* = 56) patients. The tool measures attention, executive function, memory (immediate and delayed recall), and mental flexibility using tasks such as Stroop, digit symbol substitution, and recall testing [85,86]. Scores were reported as population percentiles (1–100), with higher scores indicating better performance.

### 4.8. Statistical Analysis

Data analyses were performed using GraphPad Prism (version 8.0; GraphPad Software, Boston, MA, USA). Continuous variables are presented as mean ± standard error of the mean (SEM), and group comparisons were conducted using one-way ANOVA followed by Tukey’s post hoc test. Pearson correlation coefficients (r) were used to assess the relationships between plasma FGF-21 levels and questionnaire and cognitive scores within diagnostic and stratified subgroups. To account for multiple comparisons, the Benjamini–Hochberg procedure for false discovery rate (FDR) control was applied independently to *p*-values within each correlation table. Both raw and FDR-adjusted *p*-values are reported. A *p*-value of <0.05 was considered statistically significant. Significant correlations are indicated by * *p* < 0.05 and ** *p* < 0.01.

## 5. Conclusions

This study conclusively establishes circulating FGF-21 as a promising and multifaceted biomarker for ME and FM. We demonstrate that FGF-21 is consistently elevated in ME and ME + FM patients, with this elevation occurring independently of biological sex. Crucially, by systematically stratifying patients based on FGF-21 levels, our findings reveal distinct, phenotype-specific correlations: lower FGF-21 levels are associated with more severe symptoms (e.g., PEM in FM, autonomic/immune in ME, cognitive in ME + FM), while higher FGF-21 levels exhibit complex, sometimes paradoxical, associations with specific fatigue types and cognitive functions (e.g., improved cognition in ME only). These compelling results highlight FGF-21’s significant capacity to delineate patient subgroups beyond traditional diagnostic labels. Integrating FGF-21 into broader biomarker panels will be vital for refining disease subtyping and guiding personalized diagnostic and therapeutic strategies. Future longitudinal and interventional trials are imperative to determine if targeting FGF-21 signaling can lead to improvements in ME and FM.

## Figures and Tables

**Figure 1 ijms-26-07670-f001:**
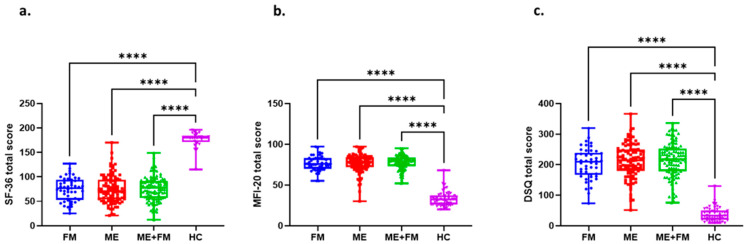
Comparison of health-related questionnaire scores across study groups. Box plots illustrating the distribution of scores for the 36-Item Short Form Survey (SF-36), the Multidimensional Fatigue Inventory (MFI-20), and the DePaul Symptom Questionnaire (DSQ) across the fibromyalgia (FM, *n* = 47), myalgic encephalomyelitis (ME, *n* = 99), myalgic encephalomyelitis with fibromyalgia (ME + FM, *n* = 104), and healthy control (HC, *n* = 54) cohorts. (**a**) For the SF-36, higher scores indicate better-perceived health, whereas higher scores indicate greater symptom severity for MFI-20 (**b**) and DSQ (**c**). Each box plot represents the distribution of scores, with the embedded box plot showing the median (central line), interquartile range (box edges), and 1.5× interquartile range (whiskers). Individual data points are overlaid as dots. Statistical significance between groups was assessed using ANOVA followed by Tukey’s post hoc test. **** *p* < 0.0001 for all comparisons between disease groups and HC groups.

**Figure 2 ijms-26-07670-f002:**
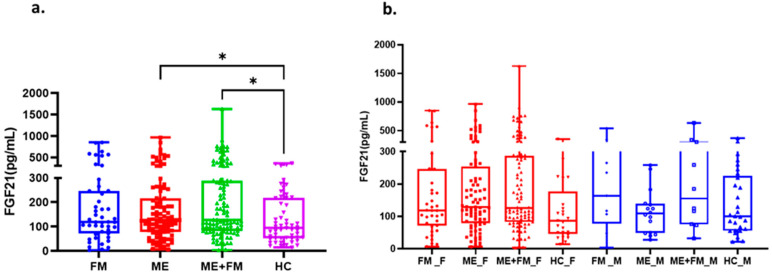
Circulating FGF-21 levels in patient groups and the impact of biological sex. (**a**) FGF-21 levels are evaluated by diagnostic group. Box plots show FGF-21 concentrations (pg/mL) on the *y*-axis for patients with fibromyalgia (FM, blue, *n* = 47), myalgic encephalomyelitis (ME, red, *n* = 99), myalgic encephalomyelitis with fibromyalgia (ME + FM, green, *n* = 104), and healthy controls (HC, purple, *n* = 54). Compared to healthy controls, higher FGF-21 levels are observed in ME and ME + FM patient groups, though significant heterogeneity exists within all cohorts. (**b**) FGF-21 levels are evaluated by biological sex. The data with females (F, red) and males (M, blue) show no statistically significant differences in FGF-21 levels among the patient subgroups (FM, ME, ME + FM) or within the healthy control group. For both panels, each box plot represents the probability density of the data, with the embedded box plot indicating the median (central line), interquartile range (box edges), and 1.5× interquartile range (whiskers). Individual data points are overlaid as dots. Statistical significance was assessed using ANOVA followed by Tukey’s post hoc test. * *p* < 0.05 is considered statistically significant.

**Figure 3 ijms-26-07670-f003:**
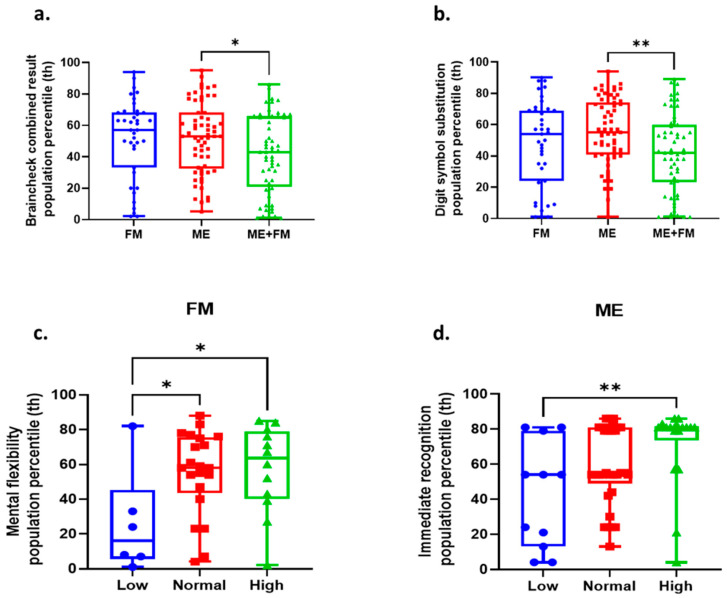
Cognitive performance and the impact of FGF-21 stratification across patient phenotypes. (**a**) Comparison of overall combined BrainCheck cognitive scores (as population percentiles) for fibromyalgia (FM, blue, *n* = 39), myalgic encephalomyelitis (ME, red, *n* = 62), and myalgic encephalomyelitis with fibromyalgia (ME + FM, green, *n* = 56) groups. (**b**) Comparison of digit symbol substitution scores (assuming higher values denote better performance) across the FM (blue, *n* = 39), ME (red, *n* = 62), and ME + FM (green, *n* = 56) groups. (**c**) Examination of mental flexibility within the FM group, stratified by FGF-21 levels (pg/mL) (assuming blue = low FGF-21 (*n* = 6), red = normal FGF-21 (*n* = 21), green = high FGF-21 (*n* = 12)). (**d**) Immediate recognition in ME subgroups based on FGF-21 (assuming blue = low FGF-21 (*n* = 11), red = normal FGF-21 (*n* = 33), green = high FGF-21 (*n* = 18)). Data presented in the box plot represents the probability density of the data, with the embedded box plot indicating the median (central line), interquartile range (box edges), and 1.5× interquartile range (whiskers). Individual data points are overlaid as dots. Statistical significance was assessed using ANOVA followed by Tukey’s post hoc test. * *p* < 0.05 and ** *p* < 0.001.

**Table 1 ijms-26-07670-t001:** Demographic and clinical characteristics of study participants.

	FM	ME	ME + FM	HC
*n*	47	99	104	54
Female/Male	38/9	83/16	94/10	25/29
Age (year)	48 ± 1.7	47 ± 1.4	48 ± 1.3	47 ± 1.5
BMI (kg/m^2^)	26 ± 0.8	26 ± 0.7	26 ± 0.8	25 ± 0.6
Illness duration (years)	11 ± 1.6	12 ± 1.1	14 ± 1.3	n/a

Demographic and clinical characteristics of patients with fibromyalgia (FM), myalgic encephalomyelitis (ME), and myalgic encephalomyelitis with fibromyalgia (ME + FM), compared to healthy controls (HC). Data are presented as mean ± standard error of the mean for continuous variables (age, BMI, illness duration) and categorical variables are reported as counts. ‘*n*’ represents the number of participants in each group. BMI: body mass index; n/a: not applicable.

**Table 2 ijms-26-07670-t002:** Correlations between low plasma FGF-21 levels and symptoms in disease groups.

Severity Score vs. FGF-21	FM (*n* = 8) (0–50 pg/mL)	ME (*n* = 17) (0–50 pg/mL)	ME + FM (*n* = 10) (0–50 pg/mL)	HC (*n* = 12) (0–50 pg/mL)
Autonomic/endocrine/ immunity (DSQ)	r = −0.19 *p* (raw) = 0.65 *p* (adj) = 0.65	r = −0.52 *p* (raw) = 0.01 *p* (adj) = 0.03 *	r = −0.61 *p* (raw) = 0.037 *p* (adj) = 0.062	r = 0.48 *p* (raw) = 0.04 *p* (adj) = 0.12
Cognitive (DSQ)	r = −0.36 *p* (raw) = 0.23 *p* (adj) = 0.38	r = −0.12 *p* (raw) = 0.65 *p* (adj) = 0.64	r = −0.67 *p* (raw) = 0.01 *p* (adj) = 0.03 *	r = 0.021 *p* (raw) = 0.47 *p* (adj) = 0.94
Post-exertional malaise (PEM) (DSQ)	r = −0.90 *p* (raw) = 0.001 *p* (adj) = 0.0023 **	r = −0.001 *p* (raw) = 0.98 *p* (adj) = 0.97	r = −0.39 *p* (raw) = 0.21 *p* (adj) = 0.26	r = 0.32 *p* (raw) = 0.23 *p* (adj) = 0.31
DPEMQ	r = −0.44 *p* (raw) = 0.3 *p* (adj) = 0.38	r = −0.65 *p* (raw) = 0.01 *p* (adj) = 0.03 *	r = 0.36 *p* (raw) = 0.38 *p* (adj) = 0.38	n/a
Mental score (SF-36)	r = −0.34 *p* (raw) = 0.073 *p* (adj) = 0.41	r = −0.42 *p* (raw) = 0.098 *p* (adj) = 0.097	r = 0.68 *p* (raw) = 0.01 *p* (adj) = 0.03 *	r = 0.34 *p* (raw) = 0.27 *p* (adj) = 0.27

Pearson correlation coefficients (r), raw *p*-values (*p* raw), and Benjamini–Hochberg false discovery rate (FDR)-adjusted *p*-values (*p* adj) are presented for the relationship between FGF-21 levels (within the 0–50 pg/mL range) and various symptom severity scores. Correlations are shown for participants in the low FGF-21 subgroup across different diagnostic categories: fibromyalgia (FM, *n* = 8), myalgic encephalomyelitis (ME, *n* = 17), myalgic encephalomyelitis with fibromyalgia (ME + FM, *n* = 10), and healthy controls (HC, *n* = 12). Symptom severity was assessed using subscales from the DSQ (Autonomic/endocrine/immunity, cognitive, PEM) and the DPEMQ, as well as the mental score from the SF-36. Asterisks denote significant correlations for *p* adj: * *p*< 0.05, ** *p*< 0.01. ‘n/a’ indicates not applicable.

**Table 3 ijms-26-07670-t003:** Correlations between low plasma FGF-21 and PEM symptoms in patients.

DSQ Questionnaire	FM (*n* = 8) (0–50 pg/mL)	ME (*n* = 17) (0–50 pg/mL)	ME + FM (*n* = 10) (0–50 pg/mL)
FGF-21 vs. Fatigue/extreme tiredness	r = −0.80 *p* (raw) = 0.01 *p* (adj) = 0.02 *	r = −0.04 *p* (raw) = 0.55 *p* (adj) = 0.89	r = −0.29 *p* (raw) = 0.21 *p* (adj) = 0.42
FGF-21 vs. Dead, heavy feeling after starting to exercise	r = −0.53 *p* (raw) = 0.018 *p* (adj) = 0.18	r = −0.23 *p* (raw) = 0.05 *p* (adj) = 0.37	r = −0.10 *p* (raw) = 0.79 *p* (adj) = 0.79
FGF-21 vs. Next day soreness or fatigue after non-strenuous, everyday activities	r = −0.77 *p* (raw) = 0.02 *p* (adj) = 0.02 *	r = −0.14 *p* (raw) = 0.29 *p* (adj) = 0.59	r = −0.29 *p* (raw) = 0.26 *p* (adj) = 0.42
FGF-21 vs. Mentally tired after the slightest effort	r = −0.89 *p* (raw) = 0.001 *p* (adj) = 0.003 **	r = 0.01 *p* (raw) = 0.84 *p* (adj) = 0.96	r = −0.64 *p* (raw) = 0.01 *p* (adj) = 0.04 *
FGF-21 vs. Minimum exercise makes you physically tired	r = −0.83 *p* (raw) = 0.01 *p* (adj) = 0.012 *	r = 0.04 *p* (raw) = 0.66 *p* (adj) = 0.89	r = −0.16 *p* (raw) = 0.48 *p* (adj) = 0.65
FGF-21 vs. Physically drained or sick after mild activity	r = −0.65 *p* (raw) = 0.07 *p* (adj) = 0.08	r = 0.15 *p* (raw) = 0.22 *p* (adj) = 0.58	r = −0.48 *p* (raw) = 0.04 *p* (adj) = 0.16
FGF-21 vs. Muscle weakness	r = −0.87 *p* (raw) = 0.002 *p* (adj) = 0.01 **	r = 0.15 *p* (raw) = 0.14 *p* (adj) = 0.57	r = 0.16 *p* (raw) = 0.56 *p* (adj) = 0.65
FGF-21 vs. PEM	r = −0.90 *p* (raw) = 0.001 *p* (adj) = 0.005 **	r = −0.01 *p* (raw) = 0.98 *p* (adj) = 0.98	r = −0.39 *p* (raw) = 0.21 *p* (adj) = 0.27

Pearson correlation coefficients (r), raw *p*-values (*p* raw), and Benjamini–Hochberg false discovery rate (FDR)-adjusted *p*-values (*p* adj) are presented for the relationship between FGF-21 levels (within the low FGF-21 subgroup, 0–50 pg/mL) and various PEM-related symptoms, primarily assessed via items from the DSQ. Correlations are shown for patients with fibromyalgia (FM), myalgic encephalomyelitis (ME), and myalgic encephalomyelitis with fibromyalgia (ME + FM). Significant correlations are denoted by asterisks for *p* adj: * *p* < 0.05, ** *p* < 0.01.

**Table 4 ijms-26-07670-t004:** Correlations between low plasma FGF-21 and DPEMQ symptoms in patients.

DPEMQ Questionnaires	FM (*n* = 8) (0–50 pg/mL)	ME (*n* = 17) (0–50 pg/mL)	ME + FM (*n* = 10) (0–50 pg/mL)
FGF-21 vs. 1. Endurance/ability to perform activities	r = 0.54 *p* (raw) = 0.10 *p* (adj) = 0.27	r = −0.51 *p* (raw) = 0.05 *p* (adj) = 0.11	r = −0.66 *p* (raw) = 0.01 *p* (adj) = 0.07
FGF-21 vs. 2. Physical fatigue, mental overstimulation	r = −0.46 *p* (raw) = 0.18 *p* (adj) = 0.35	r = 0.3435 *p* (raw) = 0.24 *p* (adj) = 0.301	r = 0.3218 *p* (raw) = 0.25 *p* (adj) = 0.43
FGF-21 vs. 3. Cognitive exhaustion	r = −0.83 *p* (raw) = 0.003 *p* (adj) = 0.043 *	r = −0.7631 *p* (raw) = 0.001 *p* (adj) = 0.006 **	r = −0.6441 *p* (raw) = 0.02 *p* (adj) = 0.084
FGF-21 vs. 4. Difficulty thinking	r = −0.56 *p* (raw) = 0.07 *p* (adj) = 0.24	r = −0.62 *p* (raw) = 0.01 *p* (adj) = 0.040 *	r = −0.75 *p* (raw) = 0.002 *p* (adj) = 0.03 *
FGF-21 vs. 5. Unrestful sleep	r = 0.10 *p* (raw) = 0.79 *p* (adj) = 0.85	r = −0.52 *p* (raw) = 0.04 *p* (adj) = 0.09	r = −0.53 *p* (raw) = 0.052 *p* (adj) = 0.18
FGF-21 vs. 6. Insomnia	r = −0.18 *p* (raw) = 0.52 *p* (adj) = 0.73	r = −0.011 *p* (raw) = 0.97 *p* (adj) = 0.97	r = 0.15 *p* (raw) = 0.56 *p* (adj) = 0.71
FGF-21 vs. 7. Muscle pain	r = 0.31 *p* (raw) = 0.35 *p* (adj) = 0.54	r = 0.018 *p* (raw) = 0.89 *p* (adj) = 0.95	r = 0.49 *p* (raw) = 0.07 *p* (adj) = 0.21
FGF-21 vs. 8. Muscle weakness	r = 0.17 *p* (raw) = 0.58 *p* (adj) = 0.74	r = −0.48 *p* (raw) = 0.08 *p* (adj) = 0.13	r = −0.08 *p* (raw) = 0.78 *p* (adj) = 0.84
FGF-21 vs. 9. Pain all over the body	r = −0.09 *p* (raw) = 0.87 *p* (adj) = 0.86	r = 0.28 *p* (raw) = 0.35 *p* (adj) = 0.40	r = 0.10 *p* (raw) = 0.68 *p* (adj) = 0.80
FGF-21 vs. 10. Dizziness	r = −0.64 *p* (raw) = 0.04 *p* (adj) = 0.172	r = −0.37 *p* (raw) = 0.19 *p* (adj) = 0.26	r = −0.05 *p* (raw) = 0.89 *p* (adj) = 0.88
FGF-21 vs. 11. Flu-like symptoms	r = 0.15 *p* (raw) = 0.66 *p* (adj) = 0.77	r = −0.76 *p* (raw) = 0.001 *p* (adj) = 0.007 **	r = −0.29 *p* (raw) = 0.31 *p* (adj) = 0.48
FGF-21 vs. 12. Temperature disturbances	r = −0.71 *p* (raw) = 0.02 *p* (adj) = 0.12	r = −0.53 *p* (raw) = 0.03 *p* (adj) = 0.95	r = −0.16 *p* (raw) = 0.50 *p* (adj) = 0.70
FGF-21 vs. 13. Mental fog	r = −0.48 *p* (raw) = 0.14 *p* (adj) = 0.34	r = −0.44 *p* (raw) = 0.11 *p* (adj) = 0.17	r = −0.40 *p* (raw) = 0.14 *p* (adj) = 0.32
FGF-21 vs. DPEMQ total score	r = −0.44 *p* (raw) = 0.3 *p* (adj) = 0.38	r = −0.65 *p* (raw) = 0.01 *p* (adj) = 0.03 *	r = −0.36 *p* (raw) = 0.38 *p* (adj) = 0.38

Pearson correlation coefficients (r), raw *p*-values (*p* raw), and Benjamini–Hochberg false discovery rate (FDR)-adjusted *p*-values (*p* adj) are presented for the relationship between FGF-21 levels (within the low FGF-21 subgroup, 0–50 pg/mL) and various individual symptom items, as well as the total score, from the DPEMQ. Correlations are shown for patients with fibromyalgia (FM), myalgic encephalomyelitis (ME), and myalgic encephalomyelitis with fibromyalgia (ME + FM). Significant correlations are denoted by asterisks for *p* adj: * *p* < 0.05, ** *p* < 0.01.

**Table 5 ijms-26-07670-t005:** Correlations between plasma FGF-21 levels and cognitive performance in patients.

FGF-21 vs.	FM (*n* = 39)	ME (*n* = 62)	ME + FM (*n* = 56)
BrainCheck combined test population percentile	r = 0.12 *p* (raw) = 0.15 *p* (adj) = 0.45	r = 0.26 *p* (raw) = 0.043 *p* (adj) = 0.043 *	r = 0.08 *p* (raw) = 0.18 *p* (adj) = 0.55
Immediate recognition population percentile	r = 0.12 *p* (raw) = 0.51 *p* (adj) = 0.51	r = 0.39 *p* (raw) = 0.001 *p* (adj) = 0.002 **	r = 0.039 *p* (raw) = 0.51 *p* (adj) = 0.77
Delayed recognition population percentile	r = 0.11 *p* (raw) = 0.33 *p* (adj) = 0.49	r = 0.33 *p* (raw) = 0.006 *p* (adj) = 0.009 **	r = 0.031 *p* (raw) = 0.81 *p* (adj) = 0.81

Pearson correlation coefficients (r), raw *p*-values (*p* raw), and Benjamini–Hochberg false discovery rate (FDR)-adjusted *p*-values (*p* adj) are presented for the relationship between circulating FGF-21 levels and various cognitive assessment scores from the BrainCheck test battery. Correlations are shown for patients with fibromyalgia (FM, *n* = 39), myalgic encephalomyelitis (ME, *n* = 62), and myalgic encephalomyelitis with fibromyalgia (ME + FM, *n* = 56). Higher percentiles indicate better mental performance. Significant correlations are denoted by an asterisk for *p* adj: * *p* < 0.05, ** *p* < 0.01.

**Table 6 ijms-26-07670-t006:** Correlations between high plasma FGF-21 levels and fatigue severity in subgroups.

Severity Score vs. FGF-21	FM (*n* = 14) (>200 pg/mL)	ME (*n* = 27) (>200 pg/mL)	ME + FM (*n* = 38) (>200 pg/mL)	HC (*n* = 13) (>200 pg/mL)
Mental fatigue (MFI-20)	r = 0.13 *p* (raw) = 0.63 *p* (adj) = 0.63	r = −0.53 *p* (raw) = 0.002 *p* (adj) = 0.004 **	r = −0.02 *p* (raw) = 0.89 *p* (adj) = 0.89	r = 0.13 *p* (raw) = 0.68 *p* (adj) = 0.68
Physical fatigue (MFI-20)	r = 0.14 *p* (raw) = 0.31 *p* (adj) = 0.61	r = −0.29 *p* (raw) = 0.14 *p* (adj) = 0.14	r = 0.37 *p* (raw) = 0.01 *p* (adj) = 0.02 *	r = 0.21 *p* (raw) = 0.25 *p* (adj) = 0.50

Pearson correlation coefficients (r), raw *p*-values (*p* raw), and Benjamini–Hochberg false discovery rate (FDR)-adjusted *p*-values (*p* adj) are presented for the relationship between FGF-21 levels (within the >200 pg/mL range) and two key dimensions of fatigue from the MFI-20: mental fatigue and physical fatigue. Correlations are shown for participants in the high FGF-21 subgroup across different diagnostic categories: fibromyalgia (FM, *n* = 14), myalgic encephalomyelitis (ME, *n* = 27), myalgic encephalomyelitis with fibromyalgia (ME + FM, *n* = 38), and healthy controls (HC, *n* = 13). Asterisks denote significant correlations for *p* adj: * *p* < 0.05, ** *p* < 0.01.

**Table 7 ijms-26-07670-t007:** Summary of studies investigating FGF-21 levels in myalgic encephalomyelitis (ME), fibromyalgia (FM), and related chronic conditions.

Study/Source	Disease Context	Key Findings on FGF-21	Key Associations & Relevance to Fatigue/Cognition	Specific Method/Notes
Current study	ME, FM, ME + FM, HC	Elevated in ME & ME + FM vs. HC. FGF-21 is a biomarker that varies in concentration.	Low FGF-21: associated with worse PEM (FM), autonomic symptoms (ME), and cognitive deficits (ME + FM). High FGF-21: associated with less mental fatigue (ME) but more physical fatigue (ME + FM).	Large, well-characterized cohort; miRNA-based diagnosis; multi-modal symptom assessment including PEM provocation; FGF-21 stratification.
Domingo et al., 2021 [39]	ME	Elevated levels in ME patients vs. healthy controls.	FGF-21 is proposed as a biomarker candidate for ME.	Plasma FGF-21 by ELISA.
Hoel et al., 2021 [5]	ME-M2 subgroup (metabolic phenotype)	Higher FGF-21 in a specific ME subgroup with low serum fatty acid derivatives.	Links FGF-21 to a distinct metabolic disruption and expands the metabolic map of ME.	Subgroup analysis based on a specific metabolic profile.
Post et al., 2021 [43]	Hemodialysis patients	Higher FGF-21 was associated with markers of malnutrition and more fatigue.	FGF-21 levels reflect a state of metabolic stress and inflammation, which is common in chronic diseases.	Study in hemodialysis patients.
Dolegowska et al., 2019 [44]	Obesity, diabetes, and metabolic syndrome	Highlights the paradox of elevated endogenous FGF-21.	Elevated FGF-21 in these chronic metabolic diseases often reflects a state of FGF-21 resistance or metabolic overload.	Review article on FGF19/FGF21.
Tan et al., 2023 [51]	Cardiovascular disease (CAD), T2DM	Elevated FGF-21 predicts cardiovascular risk and diabetic complications.	Protective anti-inflammatory and antioxidative roles. High levels may reflect an overwhelmed compensatory response or resistance.	Review of FGF-21 in metabolic and cardiovascular diseases.
Falamarzi et al., 2022 [52]	Liver diseases (NAFLD, NASH)	FGF-21 regulates hepatic lipid and glucose metabolism. Therapeutic analogs are being developed.	Elevated levels are associated with liver stress and disease, but therapeutic FGF-21 analogs can still be beneficial.	Review of FGF-21 in liver diseases.
Leng et al., 2015 [53]	Neuronal cells (in vitro)	Induces complete neuronal protection against glutamate toxicity.	Acts via Akt-1 activation and GSK-3 inhibition, showing a key neuroprotective role.	In vitro study on neuronal culture.
Yang et al., 2023 [54]	Parkinson’s disease mouse model	Preserves neurons and improves motor/cognitive scores.	Modulates gut microbiota and metabolic homeostasis, linking FGF-21 to brain–gut axis and neuroprotection.	In vivo mouse model study.
Shen et al., 2024 [55]	Parkinson’s disease cellular models	Prevents dopaminergic neuron loss.	Shows beneficial effects against proteasome impairment-induced Parkinson’s disease syndrome.	Cellular models.
Zhang et al., 2024 [56]	Diabetes-induced cognitive decline	Remodels cerebral glucose metabolism and alleviates cognitive decline.	Activation of the PI3K/AKT/GSK-3β signaling pathway.	In vivo mouse model of diabetic cognitive decline.
Wang et al., 2025 [57]	Ischemic brain injury mouse model	Suppresses astrocyte activation and protects brain tissue.	Acts through anti-inflammatory and neurotrophic pathways, showing a protective role in neuroinflammation.	In vivo mouse model study.

## Data Availability

The original contributions presented in this study are included in the article. Further inquiries can be directed to the corresponding author.

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
