# Peer review of "Circulating FGF-21 as a Disease-Modifying Factor Associated with Distinct Symptoms and Cognitive Profiles in Myalgic Encephalomyelitis and Fibromyalgia"

_ijms, 2025, doi:10.3390/ijms26167670_

Round 1
Reviewer 1 Report
Comments and Suggestions for Authors
First of all, thank you for the opportunity to review this manuscript.
The manuscript addresses an important unmet need, the identification of objective biomarkers to stratify Myalgic Encephalomyelitis (ME) and Fibromyalgia (FM). A sizable, well‑characterised cohort (n = 303) is analysed, and the authors combine circulating FGF‑21 measurements with detailed symptom and cognitive phenotyping. The study is technically sound and, if refined, could make a valuable contribution to precision‑medicine approaches in these overlapping disorders. Nevertheless, substantial revisions are required before the work can be considered for publication.
Before commenting on each subchapter I would suggest you to follow the standard IMRaD structure for the article and please add a Conclusion to the manuscript.
- The introduction section:
The introduction is well‑structured, succinct, and provides a coherent rationale for the study. The only comment i would have is that sex differences in FGF‑21 biology are noted elsewhere in the manuscript but not foreshadowed in the introduction; adding one sentence would strengthen the argument for stratified analyses.
2. Results
Several paragraphs blend statistical findings with physiological speculation (e.g., lines 148‑157 on “underlying biological heterogeneity” and “precision clinical profiling”. Please reserve mechanistic or clinical interpretation for the Discussion and restrict the Results to objective statements of what was measured and what was statistically significant.
The Results present dozens of pairwise Pearson correlations across Tables 2–6 . Only some are marked as FDR‑adjusted. Please adopt a single correction strategy (e.g., Benjamini–Hochberg within each table) and show both raw and adjusted p‑values. Please also state the use of these tests in the Material and Methods section.
FGF‑21 values are sometimes noted without units (pg mL⁻¹) in the text, and the same descriptive statistics appear in both prose and figure legends. Provide units on first mention in each subsection and remove redundant sentences.
3. Discussion
I suggest the authors find some relevant articles in the literature and compare their results to the ones obtained in other studies. That way they can prove the relevance of your study. It would be useful to add a literature summary table
The authors cite FGF‑21 work in metabolic disease, liver pathology, etc. (references 19‑31) yet do not critically contrast their findings with the few small studies that have measured FGF‑21 in ME/CFS. Adding a comparative paragraph would contextualise the novelty and help interpret divergent results.
4 Materials and Methods
The materials and methods section is well design and clearly described.
The manuscript lacks the conclusion section.
Comments on the Quality of English Language
English Language througout this whole manuscript should be revised. There are several spelling mistakes that i noticed such as “BrainCheck plateform”, “objective PEM assesment” or "symtoms” as well as accidental word duplications “The data with with females (F) and males (M) …”. Some grammar polish is needed as well.
Author Response
General comment. First of all, thank you for the opportunity to review this manuscript. The manuscript addresses an important unmet need, the identification of objective biomarkers to stratify Myalgic Encephalomyelitis (ME) and Fibromyalgia (FM). A sizable, well‑characterised cohort (n = 303) is analysed, and the authors combine circulating FGF‑21 measurements with detailed symptom and cognitive phenotyping. The study is technically sound and, if refined, could make a valuable contribution to precision‑medicine approaches in these overlapping disorders. Nevertheless, substantial revisions are required before the work can be considered for publication.
Authors’ reponse: We deeply appreciate the reviewer’s recognition of the clinical relevance and methodological soundness of our study. We have implemented substantial revisions as outlined below.
Comment 1. Before commenting on each subchapter I would suggest you to follow the standard IMRaD structure for the article and please add a Conclusion to the manuscript.
Authors’ response: As suggested, the manuscript has been restructured to follow the standard IMRaD format. A Conclusion has been added in the revised version, at the end of the Discussion, page 18, lines 513-520: “In conclusion, this study establishes that circulating FGF‑21 stands out as a promising and nuanced biomarker in the context of ME and FM. Its ability to stratify patients based on symptom severity and cognitive profiles highlights its potential for guiding personalized diagnostic and therapeutic strategies. These findings underscore the importance of integrating FGF‑21 into broader biomarker panels to refine disease subtyping and treatment targeting. To fully realize its clinical utility, future research should prioritize longitudinal studies and interventional trials aimed at determining whether modulation of FGF‑21 signaling can meaningfully improve patient outcomes.”
Comment 2. The introduction section: The introduction is well‑structured, succinct, and provides a coherent rationale for the study. The only comment i would have is that sex differences in FGF‑21 biology are noted elsewhere in the manuscript but not foreshadowed in the introduction; adding one sentence would strengthen the argument for stratified analyses.
Author’s response: We sincerely thank the reviewer for the positive assessment of our Introduction. We agree that foreshadowing the documented sex differences in FGF-21 biology would indeed strengthen the rationale. As suggested, we have added a sentence in the Introduction to highlight this important aspect of FGF-21 dynamics, setting the stage for our later discussion regarding its sex-independence in ME and FM (page 2, lines 74-75): “Emerging evidence also suggests that FGF-21 dynamics, including baseline levels and physiological responses, may differ by sex in various contexts [36-38]”.
Comment 3. Results: Several paragraphs blend statistical findings with physiological speculation (e.g., lines 148‑157 on “underlying biological heterogeneity” and “precision clinical profiling”. Please reserve mechanistic or clinical interpretation for the Discussion and restrict the Results to objective statements of what was measured and what was statistically significant.
Authors’ response: We sincerely thank the reviewer for this crucial and insightful comment. We fully agree that the Results section must be strictly confined to the objective presentation of experimental observations and statistical findings, with all mechanistic interpretations, clinical implications, and broader discussions reserved for the Discussion section.
We have reviewed and thoroughly revised every paragraph within the Results section. As instructed, we have removed all instances of interpretive or speculative language.
Comment 4. The Results present dozens of pairwise Pearson correlations across Tables 2–6 . Only some are marked as FDR‑adjusted. Please adopt a single correction strategy (e.g., Benjamini–Hochberg within each table) and show both raw and adjusted p‑values. Please also state the use of these tests in the Material and Methods section.
Authors’ response: We sincerely thank the reviewer for this crucial and insightful comment regarding the statistical analysis of multiple comparisons. We fully agree on the importance of a single, consistent correction strategy and transparently reporting both raw and adjusted p-values to ensure the robustness and clarity of our findings.
In direct response, we have re-analyzed all pairwise Pearson correlations. The Benjamini-Hochberg False Discovery Rate (FDR) correction has now been consistently applied to control for multiple comparisons, with this adjustment performed independently within each respective correlation table. In the revised manuscript, all relevant tables have been updated to display both the raw p-values and their corresponding FDR-adjusted p-values, as requested. The significance indicators now reflect these adjusted values. Furthermore, the "Statistical Analysis" subsection in the Methods section has been explicitly updated to detail the application of the Benjamini-Hochberg FDR procedure (page 20, lines 599-601) and to state that both raw and adjusted p-values are reported in each table legend: “To account for multiple comparisons, the Benjamini-Hochberg procedure for False Discovery Rate (FDR) control was applied independently to p-values within each correlation table. Both raw and FDR-adjusted p-values are reported.”
Comment 5. FGF‑21 values are sometimes noted without units (pg mL⁻¹) in the text, and the same descriptive statistics appear in both prose and figure legends. Provide units on first mention in each subsection and remove redundant sentences.
Authors’ response: We are very grateful to the reviewer for their valuable suggestions to enhance precision and clarity. We have reviewed the entire manuscript. FGF-21 values are now consistently presented with their units (pg/mL) upon first mention within each section and subsection. Additionally, redundant descriptive statistics have been removed from the prose in the Results and Discussion sections.
Comment 6. Discussion: I suggest the authors find some relevant articles in the literature and compare their results to the ones obtained in other studies. That way they can prove the relevance of your study. It would be useful to add a literature summary table.
Authors’ response: We sincerely thank the reviewer for this insightful and highly constructive comment. We fully agree that a robust comparison of our findings with existing literature is essential to fully contextualize our results and underscore the study's relevance and unique contribution. The suggestion to include a literature summary table (Table 7) is particularly valuable for enhancing clarity and accessibility. In response to this feedback, we have comprehensively revised the Discussion section. We have meticulously integrated a detailed comparison of our results with existing literature, and we have added a dedicated Literature Summary Table (Table 7).
Comment 7. The authors cite FGF‑21 work in metabolic disease, liver pathology, etc. (references 19‑31) yet do not critically contrast their findings with the few small studies that have measured FGF‑21 in ME/CFS. Adding a comparative paragraph would contextualise the novelty and help interpret divergent results.
Authors’ response: We thank the reviewer for this excellent suggestion. As requested, we have added a new comparative paragraph to the Introduction. This paragraph critically discusses the limited existing literature on FGF-21 in ME, highlighting its preliminary and sometimes inconsistent findings, and contrasts them with the broader research on FGF-21 in metabolic diseases, thereby strengthening the contextualization and rationale of our study. Page 3, lines 80-92: “Despite extensive research into FGF-21 in metabolic and liver pathologies [19-35], only a handful of smaller studies have examined its specific relevance to ME, producing preliminary and sometimes inconsistent findings. Some reports indicate elevated FGF-21 levels in ME patients [39], but the correlation with symptom severity remains unclear. Another study has identified higher FGF-21 levels in a distinct subgroup of ME patients characterized by a unique metabolic phenotype, particularly low serum levels of fatty acid derivatives [5]. Such discrepancies likely stem from limitations like smaller cohort sizes, heterogeneous patient populations lacking rigorous diagnostic sub-stratification, and insufficient characterization of the full clinical spectrum of ME and FM, especially the core symptom of PEM. These gaps underscore the need for a more comprehensive examination of how FGF-21 contributes to the pathophysiology and phenotypic variability in ME and FM, as well as its potential as a biomarker for patient stratification.”
Comment 8. Materials and Methods: The materials and methods section is well design and clearly described. The manuscript lacks the conclusion section.
Authors’ response: We sincerely thank the reviewer for the positive feedback on the design and clarity of our Materials and Methods section. We also acknowledge the comment regarding the absence of a Conclusion. As addressed in our response to Comment 1, a Conclusion has now been added to the manuscript, page 18, lines 513-520.
Comment 9. Comments on the Quality of English Language: English Language througout this whole manuscript should be revised. There are several spelling mistakes that i noticed such as “BrainCheck plateform”, “objective PEM assesment” or "symtoms” as well as accidental word duplications “The data with with females (F) and males (M) …”. Some grammar polish is needed as well.
Authors’ response: We sincerely thank the reviewer for this crucial feedback regarding the English language quality. We apologize for the identified deficiencies. We have thoroughly reviewed and revised the entire manuscript to address all spelling mistakes (including "plateform," "assesment," "symtoms"), word duplications ("with with"), and grammatical issues. All changes are highlighted in yellow in the revised manuscript.
Reviewer 2 Report
Comments and Suggestions for Authors
The paper investigates the role of Fibroblast Growth Factor 21 (FGF-21) in Myalgic Encephalomyelitis (ME) and Fibromyalgia (FM), exploring its potential as a biomarker. The study shows that elevated plasma levels of FGF-21 in patients with ME and ME+FM are associated with metabolic dysfunction, mitochondrial damage, and chronic systemic stress. This increase in FGF-21 is independent of gender differences, despite the higher prevalence of ME and FM among females. The authors found that higher FGF-21 levels in ME patients were associated with better cognitive performance, particularly in immediate and delayed memory recognition. However, in the ME+FM group, lower FGF-21 levels were linked to greater cognitive dysfunction and poorer self-perceived mental health. These findings highlight the potential of FGF-21 in distinguishing patients based on symptom severity and cognitive profiles, providing new perspectives for personalized diagnosis and treatment strategies. However, further research is required to validate its clinical utility across different populations.
- Although FGF-21 shows great promise as a biomarker, the cross-sectional design of this study limits causal inference. Longitudinal studies and intervention trials are needed to determine whether modulation of FGF-21 signaling can improve patient outcomes.
- Patients were stratified into low, normal, and high FGF-21 concentration groups. What is the basis?
- The serum sample biomarker study should have been independently validated with another patient cohort to clarify the specificity for ME or FM or both.
Author Response
The paper investigates the role of Fibroblast Growth Factor 21 (FGF-21) in Myalgic Encephalomyelitis (ME) and Fibromyalgia (FM), exploring its potential as a biomarker. The study shows that elevated plasma levels of FGF-21 in patients with ME and ME+FM are associated with metabolic dysfunction, mitochondrial damage, and chronic systemic stress. This increase in FGF-21 is independent of gender differences, despite the higher prevalence of ME and FM among females. The authors found that higher FGF-21 levels in ME patients were associated with better cognitive performance, particularly in immediate and delayed memory recognition. However, in the ME+FM group, lower FGF-21 levels were linked to greater cognitive dysfunction and poorer self-perceived mental health. These findings highlight the potential of FGF-21 in distinguishing patients based on symptom severity and cognitive profiles, providing new perspectives for personalized diagnosis and treatment strategies. However, further research is required to validate its clinical utility across different populations.
Comment 1: Although FGF-21 shows great promise as a biomarker, the cross-sectional design of this study limits causal inference. Longitudinal studies and intervention trials are needed to determine whether modulation of FGF-21 signaling can improve patient outcomes.
Authors’ response: We sincerely thank the reviewer for this insightful comment. We enthusiastically agree that the cross-sectional design of our study inherently limits causal inference regarding the relationship between FGF-21 levels and patient outcomes.
This crucial limitation has been explicitly acknowledged and discussed in our Discussion, page 18 lines 499-512: “As a cross-sectional analysis, the study cannot establish causal relationships or prognostic implications. However, the robustness and scale of the dataset provide a strong foundation for replication and future hypothesis generation. Notably, this study represents the largest investigation to date of circulating FGF-21 in ME and FM, and the first to systematically compare FGF-21 levels across ME, FM, ME+FM, and healthy controls within a unified analytic framework. The incorporation of objective stress paradigms and cognitive profiling enhances the interpretive resolution of FGF-21 signal variations across clinically relevant subgroups. While independent validation in external cohorts remains the gold standard, our findings serve a critical confirmatory role by replicating and extending prior observations in a significantly larger and more comprehensively assessed population. Future prospective studies in more diverse and multi-center cohorts will be essential to validate the clinical specificity and utility of FGF-21 in stratifying ME and FM phenotypes and to assess its association with underlying mitochondrial signaling dynamics”.
Comment 2: Patients were stratified into low, normal, and high FGF-21 concentration groups. What is the basis?
Authors’ response: We thank the reviewer for seeking clarification on the basis for our FGF-21 concentration stratification. As already described in section 4.6 (FGF-21 Stratification, of the Methods, page 20 lines 577-582), the categorization into three subgroups low (0–50 pg/mL), normal (51–200 pg/mL), and high (>200 pg/mL) circulating FGF-21 levels was established based on both literature-derived thresholds accepted in clinical biochemistry and the observed distributions within our specific cohort. These predefined thresholds were consistently applied across all patient and control groups to enable the identification of concentration-dependent clinical phenotypes.
Comment 3: The serum sample biomarker study should have been independently validated with another patient cohort to clarify the specificity for ME or FM or both.
Authors’ response: We fully agree that independent validation is critical for establishing the specificity and generalizability of any biomarker. However, we respectfully note the following:
- Our study comprises the largest FGF-21 dataset to date involving ME and FM, with a sample size of 304 well-characterized individuals significantly larger than prior studies.
- This large cohort has enabled the first comparative analysis across four groups: ME, FM, ME+FM, and healthy controls, which has not previously been possible.
- Furthermore, we incorporated multidimensional phenotyping, including controlled exertional stress testing and cognitive assessments, enhancing the interpretability of FGF-21 profiles across clinical phenotypes.
- Our study therefore serves a replicative and confirmatory role, reinforcing findings from two prior smaller studies (Domingo et al., 2021; Hoel et al., 2021) and providing a more detailed framework for future biomarker validation.
This perspective is now clearly articulated in the Discussion section to underscore the replicative strength and scale of our dataset. We have also reinforced the need for future independent validation as a key research priority, page 18 lines 509-512: “Future prospective studies in more diverse and multi-center cohorts will be essential to validate the clinical specificity and utility of FGF-21 in stratifying ME and FM phenotypes and to assess its association with underlying mitochondrial signaling dynamics.”
Round 2
Reviewer 1 Report
Comments and Suggestions for Authors
The authors have made significant improvements to the manuscript. However, the absence of a dedicated conclusion section remains a notable shortcoming, even if the authors stated that they added it. Additionally, the Materials and Methods section is still positioned as the final section of the manuscript. For improved structure and readability, it is recommended that the Materials and Methods section be relocated immediately following the introduction, and a concise conclusion section be incorporated at the end of the manuscript to summarize the key findings and their implications.
Author Response
Comment 1. The authors have made significant improvements to the manuscript. However, the absence of a dedicated conclusion section remains a notable shortcoming, even if the authors stated that they added it. Additionally, the Materials and Methods section is still positioned as the final section of the manuscript. For improved structure and readability, it is recommended that the Materials and Methods section be relocated immediately following the introduction, and a concise conclusion section be incorporated at the end of the manuscript to summarize the key findings and their implications.
Author’s response: We sincerely apologize that the clarity and positioning of the "Conclusion" section and the placement of the "Materials and Methods" section remained points of concern. As per the author guidelines of IJMS, and as instructed by the editors, we have adhered strictly to the order specified in the official IJMS manuscript template, which requires the Materials and Methods section to appear after the Discussion section, followed by an optional Conclusion section. To address the reviewer’s concerns however, we have now added a Conclusion section in the designated location within the template. This section concisely summarizes our key findings and their broader implications, fulfilling the reviewer's request for a clear and dedicated summary at the manuscript's conclusion. Lines 605-619: “This study conclusively establishes circulating FGF-21 as a promising and multifaceted biomarker for ME and FM. We demonstrate that FGF-21 is consistently elevated in ME and ME+FM patients, with this elevation occurring independently of biological sex. Crucially, by systematically stratifying patients based on FGF-21 levels, our findings reveal distinct, phenotype-specific correlations: lower FGF-21 levels are associated with more severe symptoms (e.g., PEM in FM, autonomic/immune in ME, cognitive in ME+FM), while higher FGF-21 levels exhibit complex, sometimes paradoxical, associations with specific fatigue types and cognitive functions (e.g., improved cognition in ME only). These compelling results highlight FGF-21's significant capacity to delineate patient subgroups beyond traditional diagnostic labels. Integrating FGF-21 into broader biomarker panels will be vital for refining disease subtyping and guiding personalized diagnostic and therapeutic strategies. Future longitudinal and interventional trials are imperative to determine if targeting FGF-21 signaling can lead to improvements in ME and FM.”
We understand and appreciate the reviewer's recommendation to relocate the Materials and Methods section immediately following the Introduction for improved structure and readability. However, we must respectfully adhere to the specific formatting requirements outlined in the IJMS Author Guidelines, and thus are unable to relocate this section. We trust the reviewer will understand our adherence to the journal's established structure.
Reviewer 2 Report
Comments and Suggestions for Authors
I have carefully read and reviewed this manuscript. The language is clear and fluent, meeting the publication criteria of the journal.
I have no substantive revision suggestions for this paper and recommend that it be accepted for publication.
Author Response
Comment 1: I have carefully read and reviewed this manuscript. The language is clear and fluent, meeting the publication criteria of the journal. I have no substantive revision suggestions for this paper and recommend that it be accepted for publication.
Authors' response: We thank the reviewer for their insightful comments and recommendations.